



# Idealized Models of the Joint Probability Distribution of Wind Speeds

Adam H. Monahan

School of Earth and Ocean Sciences, University of Victoria, P.O. Box 3065 STN CSC, Victoria, BC, Canada, V8W 3V6

*Correspondence to:* Adam Monahan (monahana@uvic.ca)

**Abstract.** The joint probability distribution of wind speeds at two separate locations in space or points in time completely characterizes the statistical dependence of these two quantities, providing more information than linear measures such as correlation. In this study, we consider two models of the joint distribution of wind speeds obtained from idealized models of the dependence structure of the horizontal wind velocity components. The bivariate Rice distribution follows from assuming that the wind components have Gaussian and isotropic fluctuations. The bivariate Weibull distribution arises from power law transformations of wind speeds corresponding to vector components with Gaussian, isotropic, mean-zero variability. Maximum likelihood estimates of these distributions are compared using wind speed data from the mid-troposphere, from different altitudes at the Cabauw tower in the Netherlands, and from scatterometer observations over the sea surface. While the bivariate Rice distribution is more flexible and can represent a broader class of dependence structures, the bivariate Weibull distribution is mathematically simpler and may be more convenient in many applications. The complexity of the mathematical expressions obtained for the joint distributions suggests that the development of explicit functional forms for multivariate speed distributions from distributions of the components will not be practical for more complicated dependence structure or more than two speed variables.



# 1   Introduction

A fundamental issue in the characterization of atmospheric variability is that of dependence: how the state of one atmospheric variable is related to that of another at a different position in space, or point in time. The simplest measure of statistical dependence, the correlation coefficient, is a natural measure for Gaussian-distributed quantities but does not fully characterize dependence for non-Gaussian variables. The most general representation of dependence between two or more quantities is their joint probability distribution. The joint probability distribution for a multivariate Gaussian is well-known, and expressed in terms of the mean and covariance matrix (e.g. Wilks, 2005; von Storch and Zwiers, 1999). No such general expressions for non-Gaussian multivariate distributions exist. Copula theory (e.g. Schlözel and Friederichs, 2008) allows joint distributions to be constructed from specified marginal distributions. However, which copula model to use for a given analysis is not generally known *a priori* and is usually determined empirically through a statistical model selection exercise.

The present study considers the bivariate joint probability distribution of wind speeds. As these are quantities which are by definition bounded below by zero, the joint distribution and the marginal distributions are non-Gaussian. While previous studies have considered the correlation structure (or equivalently, the power spectrum) of wind speeds in time (e.g. Brown et al., 1984; Schlax et al., 2001; Gille, 2005; Monahan, 2012b) and in space (e.g. Carlin and Haslett, 1982; Nastrom and Gage, 1985; Wylie et al., 1985; Brown and Swail, 1988; Xu et al., 2011) relatively little attention has been paid to developing expressions for the joint distribution. Copula methods have been used to model spatial dependence of wind speeds for wind power applications (Grothe and Schnieders, 2011; Louie, 2012; Veeramachaneni et al., 2015) and dependence of daily wind speed maxima (Schlözel and Friederichs, 2008). A concrete example of an analysis in which the need for an explicit parametric form for the joint distribution of wind speeds has arisen is the Hidden Markov Model (HMM) analysis considered in Monahan et al. (2015). In an HMM analysis of continuous variables, it is necessary to specify the parametric form of the joint distribution within each hidden state. In Monahan et al. (2015), the joint distribution of wind speeds at 10 m and 200 m and the potential temperature difference between these altitudes was modelled as a multivariate Gaussian distribution, although for at least the speeds this distribution cannot be correct. This pragmatic modelling approximation was made because of the absence of a more appropriate parametric distribution for the quantities being considered. The alternative approach of using the wind components at the two levels (for which the multivariate Gaussian model may be a better approximation) instead of the speeds directly has the downside of increasing the dimensionality of the state vector from three to five, dramatically increasing the number of parameters to be estimated (with the covariance matrices in particular increasing from nine to twenty-five elements) and reducing the statistical robustness of the results.

A number of previous studies have constructed univariate speed distributions starting from models for the joint distribution of the horizontal components (e.g. Cakmur et al., 2004; Monahan, 2007; Drobinski et al., 2015). One useful benefit of this approach is that it allows the statistics of the speed and the components to be related to each other. The specific goal of the present study is to extend this approach to bivariate distributions, constructing models of the joint probability distribution of wind speeds that are directly connected to the joint distributions of the horizontal components of the wind. As the following results will demonstrate, generalizing this approach to the bivariate speed distribution results in rather complicated mathematical



expressions. Expressions for multivariate distributions of more than two speeds will be even more complicated, and may not even be analytically tractable. Through the analysis of the bivariate speed distribution, we will probe how far the development of closed-form, analytic expressions for parametric speed distributions based on distributions of components can practically be extended.

Both Weibull and Rice distributions have been used to model the univariate wind speed distribution (c.f. Carta et al., 2009; Monahan, 2014; Drobinski et al., 2015), and models of multivariate distributions with Ricean or Weibull marginals have been developed (e.g. Crowder, 1989; Lu and Bhattacharyya, 1990; Kotz et al., 2000; Sagias and Karagiannidis, 2005; Yacoub et al., 2005; Mendes and Yacoub, 2007; Villanueva et al., 2013). Much of this work has been done in the context of wireless communications (Sagias and Karagiannidis, 2005; Yacoub et al., 2005; Mendes and Yacoub, 2007): the present study builds upon the
results of these earlier ones.

The two probability density functions (pdfs) we will consider, the bivariate Rice and Weibull distributions, both start with simple assumptions regarding the distributions of the wind components. The bivariate Rice distribution follows directly from the assumption of Gaussian components with isotropic variability, but nonzero mean. In contrast, the bivariate Weibull distribution is obtained from nonlinear transformations of the magnitudes of Gaussian, isotropic, mean-zero components. While
the univariate Weibull distribution has been found to generally be a better fit to observed wind speeds than the univariate Rice distribution (particularly over the oceans, e.g. Monahan, 2006, 2007), the direct connection of the Rice distribution to the distribution of the components (which the Weibull distribution does not have) is useful from a modelling and theoretical perspective (e.g. Cakmur et al., 2004; Monahan, 2012a; Culver and Monahan, 2013; Sun and Monahan, 2013; Drobinski et al., 2015). The six-parameter bivariate Rice distribution that we will consider is more flexible than the five-parameter bivariate Weibull dis-
tribution, and able to model a broader range of dependence structures. Furthermore, it is directly connected to the univariate distributions and dependence structure of the wind components. However, the bivariate Weibull distribution is mathematically much simpler than the bivariate Rice distribution and easier to use in practice.

The bivariate Rice and Weibull distributions are developed in Section 2, starting from discussion of the bivariate Rayleigh distribution (which is a limiting case of both of the other models). In this section, we repeat some of the formulae obtained
by Sagias and Karagiannidis (2005), Yacoub et al. (2005), and Mendes and Yacoub (2007) for completeness and because of notational differences between this study and the earlier ones. In Section 3, the ability of these distributions to model wind speed data from the middle troposphere, and from the near-surface over land and the ocean is considered. Examples of dependence structures in both space (horizontally and vertically) and in time are considered. A discussion and conclusions are presented in Section 4.

**2   Empirical models of the bivariate wind speed distribution**

As a starting point for developing models of the bivariate wind speed distribution, we consider the joint distribution of the horizontal wind vector components $\boldsymbol{u}_i = (u_i, v_i)$, $i = 1, 2$ (where the subscripts $i$ denote wind vectors at two different locations, two different points in time, or both). In particular, we assume that



1: the two orthogonal wind components are marginally Gaussian with isotropic and uncorrelated fluctuations:

$$\begin{pmatrix} u_i \\ v_i \end{pmatrix} \sim \mathcal{N}\left[ \begin{pmatrix} \overline{u}_i \\ \overline{v}_i \end{pmatrix}, \begin{pmatrix} \sigma_i^2 & 0 \\ 0 & \sigma_i^2 \end{pmatrix} \right] \quad i = 1, 2, \tag{1}$$

2: the cross correlation matrix of the two vectors is

$$\mathrm{corr}(\boldsymbol{u}_1, \boldsymbol{u}_2) = \begin{pmatrix} \mathrm{corr}(u_1, u_2) & \mathrm{corr}(u_1, v_2) \\ \mathrm{corr}(v_1, u_2) & \mathrm{corr}(v_1, v_2) \end{pmatrix} = \begin{pmatrix} \mu_1 & \mu_2 \\ -\mu_2 & \mu_1 \end{pmatrix} = \begin{pmatrix} \rho\cos\psi & \rho\sin\psi \\ -\rho\sin\psi & \rho\cos\psi \end{pmatrix}, \tag{2}$$

where we have expressed the correlations in both Cartesian and polar coordinates: $(\mu_1, \mu_2) = (\rho\cos\psi, \rho\sin\psi)$, with $0 \le \rho \le 1$.

Positive-definiteness of the covariance matrix of the vector $(u_1, v_1, u_2, v_2)$ requires that $\mu_1^2 + \mu_2^2 = \rho^2 \le 1$. The assumed correlation structure implies that the correlation matrix becomes diagonal when the vector $\boldsymbol{u}_2$ is rotated through the angle $-\psi$:

$$\mathrm{corr}(\boldsymbol{u}_1, R(-\psi)(\boldsymbol{u}_2)) = \begin{pmatrix} \rho & 0 \\ 0 & \rho \end{pmatrix}, \tag{3}$$

where

$$R(-\psi)\boldsymbol{u}_2 = \begin{pmatrix} \cos\psi & \sin\psi \\ -\sin\psi & \cos\psi \end{pmatrix} \begin{pmatrix} u_2 \\ v_2 \end{pmatrix} = \begin{pmatrix} u_2\cos\psi + v_2\sin\psi \\ -u_2\sin\psi + v_2\cos\psi \end{pmatrix}. \tag{4}$$

The joint distribution of the horizontal components resulting from these assumptions is

$$p(u_1, u_2, v_1, v_2) = \frac{1}{(2\pi)^2 \sigma_1^2 \sigma_2^2 (1-\rho^2)} \exp\left( -\frac{1}{2(1-\rho^2)} \left[ \frac{(u_1-\overline{u}_1)^2}{\sigma_1^2} + \frac{(v_1-\overline{v}_1)^2}{\sigma_1^2} + \frac{(u_2-\overline{u}_2)^2}{\sigma_2^2} + \frac{(v_2-\overline{v}_2)^2}{\sigma_2^2} \right. \right.$$

$$\left. \left. -\frac{2\mu_1[(u_1-\overline{u}_1)(u_2-\overline{u}_2)+(v_1-\overline{v}_1)(v_2-\overline{v}_2)]}{\sigma_1\sigma_2} - \frac{2\mu_2[(u_1-\overline{u}_1)(v_2-\overline{v}_2)-(v_1-\overline{v}_1)(u_2-\overline{u}_2)]}{\sigma_1\sigma_2} \right] \right) \tag{5}$$

(Mendes and Yacoub, 2007).

Note that only considering the horizontal components of the wind vector implicitly restricts the resulting distributions to time scales sufficiently long that the vertical component of the wind contributes negligibly to the speed.

## 2.1 Bivariate Rayleigh Distribution

The joint distributions of the speeds $w_i = \sqrt{u_i^2 + v_i^2}$ obtained from the pdf Eq. (5) when both vector wind components are mean-zero is the bivariate Rayleigh distribution (e.g. Battjes, 1969). Transforming variables to wind speed $w_i$ and direction $\theta_i = \tan^{-1}(v_i/u_i)$, the joint distribution (Eq. 5) with $\overline{u}_i = \overline{v}_i = 0$, $i = 1, 2$, becomes

$$p(w_1, w_2, \theta_1, \theta_2) = \frac{w_1 w_2}{(2\pi)^2 \sigma_1^2 \sigma_2^2 (1-\rho^2)} \exp\left( -\frac{1}{2(1-\rho^2)} \left[ \frac{w_1^2}{\sigma_1^2} + \frac{w_2^2}{\sigma_2^2} \right] \right) \exp\left( \frac{\rho}{1-\rho^2} \frac{w_1 w_2}{\sigma_1\sigma_2} \cos(\theta_1 - \theta_2 + \psi) \right). \tag{6}$$



Integrating over the wind directions to obtain the marginal distribution for the wind speeds, we obtain

$$p(w_1, w_2) = \frac{w_1 w_2}{\sigma_1^2 \sigma_2^2 (1 - \rho^2)} \exp\left(-\frac{1}{2(1-\rho^2)}\left[\frac{w_1^2}{\sigma_1^2} + \frac{w_2^2}{\sigma_2^2}\right]\right) I_o\left(\frac{\rho}{1-\rho^2}\frac{w_1 w_2}{\sigma_1 \sigma_2}\right), \tag{7}$$

where we have used the fact that

$$\int_0^{2\pi} e^{\alpha \cos \theta} \cos k\theta \, d\theta = 2\pi I_k(\alpha), \tag{8}$$

where $I_k(z)$ is the modified Bessel function of order $k$. Note that the correlation angle $\psi$ drops out after integration over $\theta_1$ and $\theta_2$. As a result, for the bivariate Rayleigh distribution, $p(w_1, w_2)$ depends only on the three parameters $(\sigma_1, \sigma_2, \rho)$.

For $\rho = 0$, $p(w_1, w_2)$ factors as the product of the marginal distributions of $w_1$ and $w_2$:

$$p(w_1, w_2) = \left[\frac{w_1}{\sigma_1^2} \exp\left(-\frac{w_1^2}{2\sigma_1^2}\right)\right]\left[\frac{w_1}{\sigma_1^2} \exp\left(-\frac{w_1^2}{2\sigma_1^2}\right)\right], \tag{9}$$

and the wind speeds are independent. As $\rho \to 1$, we can use the asymptotic result

$$I_0(x) \sim \frac{e^x}{\sqrt{2\pi x}} \qquad (x \gg 1) \tag{10}$$

to show that

$$p(w_1, w_2) \to \frac{w_1}{\sigma_1^2} \exp\left(-\frac{w_1^2}{2\sigma_1^2}\right) \delta\left(\frac{w_1}{\sigma_1} - \frac{w_2}{\sigma_2}\right), \tag{11}$$

where $\delta(\cdot)$ is the Dirac delta function. In this limit, $w_1$ and $w_2$ are perfectly correlated and Rayleigh distributed.

Moments of the bivariate Rayleigh distribution are given by

$$\mathsf{E}\{w_1^m w_2^n\} = 2^{m/2} 2^{n/2} \sigma_1^m \sigma_2^n \Gamma\left(1 + \frac{m}{2}\right) \Gamma\left(1 + \frac{n}{2}\right) {}_2F_1\left(-\frac{m}{2}, -\frac{n}{2}, 1; \rho^2\right), \tag{12}$$

where ${}_2F_1(\alpha, \beta, \gamma; z)$ is the hypergeometric function (Gradshteyn and Ryzhik, 2000). In particular, we have

$$\mathrm{mean}(w_i) = \sigma_i \sqrt{\frac{\pi}{2}} \tag{13}$$

$$\mathrm{var}(w_i) = 2\sigma_i^2 \left(1 - \frac{\pi}{4}\right) \tag{14}$$

$$\mathrm{corr}(w_1, w_2) = \frac{\pi}{4 - \pi}\left({}_2F_1\left[-\frac{1}{2}, -\frac{1}{2}, 1; \rho^2\right] - 1\right). \tag{15}$$

Because ${}_2F_1(-1/2, -1/2, 1; \rho^2)$ is an increasing function of $\rho^2$ with ${}_2F_1(-1/2, -1/2, 1; 0) = 1$, $\mathrm{corr}(w_1, w_2)$ must be non-negative for the bivariate Rayleigh distribution.

Plots of $p(w_1, w_2)$ for the three values $\rho = 0, 0.5,$ and $0.85$ are presented in Figure 1, along with the marginal distributions for $w_1$ and $w_2$ (which are the same for all three panels). The marginal distributions are positively skewed and the contours of the joint distributions are more tightly concentrated below and to the left of their peaks than elsewhere. As expected, the

distributions become more tightly concentrated around the 1:1 line as the dependence parameter $\rho$ increases.





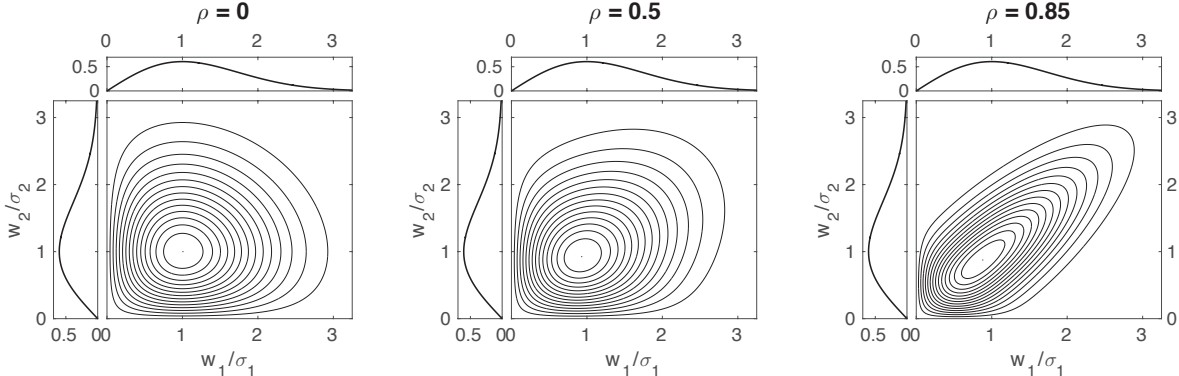

**Figure 1.** Example bivariate Rayleigh distributions $p(w_1, w_2)$ for $\rho = 0, 0.5,$ and $0.85$, with $w_1$ and $w_2$ scaled respectively by $\sigma_1$ and $\sigma_2$. The upper and left subpanels show the marginal distributions of $w_1$ and $w_2$ respectively. These marginal distributions are the same for all three panels.

## 2.2 Bivariate Rice Distribution

The assumptions leading to the bivariate Rayleigh distribution are too restrictive to model observed wind speeds in most circumstances. A more general distribution results from assuming that the wind components are Gaussian, isotropic, and uncorrelated, but with nonzero mean (Eq. 5).

Changing variables to wind speed $w_i$ and direction $\theta_i$, the joint distribution becomes

$$p(w_1, w_2, \theta_1, \theta_2) = \frac{w_1 w_2}{(2\pi)^2 \sigma_1^2 \sigma_2^2 (1-\rho^2)} \exp\left(-\frac{1}{2(1-\rho^2)}\left[\frac{(w_1^2 + \overline{u}_1^2 + \overline{v}_1^2)}{\sigma_1^2} + \frac{(w_2^2 + \overline{u}_2^2 + \overline{v}_2^2)}{\sigma_2^2} - \frac{2\mu_1(\overline{u}_1\overline{u}_2 + \overline{v}_1\overline{v}_2)}{\sigma_1\sigma_2} - \frac{2\mu_2(\overline{u}_1\overline{v}_2 - \overline{v}_1\overline{u}_2)}{\sigma_1\sigma_2}\right]\right) \times$$

$$\exp\left(\left[\frac{w_1}{\sigma_1}a_1\cos\theta_1 + \frac{w_2}{\sigma_2}a_2\cos\theta_2 + \frac{w_1}{\sigma_1}b_1\sin\theta_1 + \frac{w_2}{\sigma_2}b_2\sin\theta_2 + \frac{1}{1-\rho^2}\frac{w_1 w_2}{\sigma_1\sigma_2}\left(\mu_1\cos(\theta_1 - \theta_2) + \mu_2\sin(\theta_1 - \theta_2)\right)\right]\right), \quad (16)$$

where

$$a_1 = \frac{1}{1-\rho^2}\left(\frac{\overline{u}_1}{\sigma_1} - \mu_1\frac{\overline{u}_2}{\sigma_2} - \mu_2\frac{\overline{v}_2}{\sigma_2}\right) = \frac{1}{1-\rho^2}\left[\frac{\overline{\mathcal{U}}_1}{\sigma_1}\cos\phi_1 - \rho\frac{\overline{\mathcal{U}}_2}{\sigma_2}\cos(\phi_2 - \psi)\right] \quad (17)$$

$$b_1 = \frac{1}{1-\rho^2}\left(\frac{\overline{v}_1}{\sigma_1} - \mu_1\frac{\overline{v}_2}{\sigma_2} + \mu_2\frac{\overline{u}_2}{\sigma_2}\right) = \frac{1}{1-\rho^2}\left[\frac{\overline{\mathcal{U}}_1}{\sigma_1}\sin\phi_1 - \rho\frac{\overline{\mathcal{U}}_2}{\sigma_2}\sin(\phi_2 - \psi)\right] \quad (18)$$

$$a_2 = \frac{1}{1-\rho^2}\left(\frac{\overline{u}_2}{\sigma_2} - \mu_1\frac{\overline{u}_1}{\sigma_1} + \mu_2\frac{\overline{v}_1}{\sigma_1}\right) = \frac{1}{1-\rho^2}\left[\frac{\overline{\mathcal{U}}_2}{\sigma_2}\cos\phi_2 - \rho\frac{\overline{\mathcal{U}}_1}{\sigma_1}\cos(\phi_1 + \psi)\right] \quad (19)$$

$$b_2 = \frac{1}{1-\rho^2}\left(\frac{\overline{v}_2}{\sigma_2} - \mu_1\frac{\overline{v}_1}{\sigma_1} - \mu_2\frac{\overline{u}_1}{\sigma_1}\right) = \frac{1}{1-\rho^2}\left[\frac{\overline{\mathcal{U}}_2}{\sigma_2}\sin\phi_2 - \rho\frac{\overline{\mathcal{U}}_1}{\sigma_1}\sin(\phi_1 + \psi)\right], \quad (20)$$

and where we have defined the magnitude and direction of the mean vector wind:

$$(\overline{u}_i, \overline{v}_i) = \overline{\mathcal{U}}_i(\cos\phi_i, \sin\phi_i). \quad (21)$$




The marginal distribution for the wind speeds is obtained by integrating the joint distribution over $\theta_1$ and $\theta_2$. To evaluate this integral, we make use of the result

$$\frac{1}{(2\pi)^2} \int\limits_0^{2\pi} \int\limits_0^{2\pi} \exp\left[\alpha_1\cos\theta_1 + \alpha_2\cos\theta_2 + \beta_1\sin\theta_1 + \beta_2\sin\theta_2 + \gamma\cos(\theta_1 - \theta_2 + \psi)\right] d\theta_1 d\theta_2$$

$$= \sum_{k=0}^\infty \epsilon_k \cos\left[k\left(\tan^{-1}\frac{\beta_1}{\alpha_1} - \tan^{-1}\frac{\beta_2}{\alpha_2} + \psi\right)\right] I_k\left(\sqrt{\alpha_1^2 + \beta_1^2}\right) I_k\left(\sqrt{\alpha_2^2 + \beta_2^2}\right) I_k(\gamma), \tag{22}$$

where

$$\epsilon_k = \begin{cases} 1 & k = 0 \\ 2 & k \neq 0 \end{cases} \tag{23}$$

and it is important that $\tan^{-1}(b/a)$ be evaluated as the angle between the vector $(a,b)$ and the vector $(1,0)$ (that is, as the four-quadrant inverse tangent). Eq. (22) follows from the Fourier series

$$e^{c\cos\theta} = \sum_{k=0}^\infty \epsilon_k I_k(c) \cos k\theta \tag{24}$$

along with repeated use of trigonometric identities and the integral Eq. (8).

Finally, we obtain the expression for the bivariate Rice distribution (Mendes and Yacoub, 2007)

$$p(w_1, w_2) = \frac{w_1 w_2}{\sigma_1^2 \sigma_2^2 (1 - \rho^2)} \exp\left(-\frac{1}{2(1-\rho^2)}\left[\frac{(w_1^2 + \overline{u}_1^2 + \overline{v}_1^2)}{\sigma_1^2} + \frac{(w_2^2 + \overline{u}_2^2 + \overline{v}_2^2)}{\sigma_2^2} - \frac{2\mu_1(\overline{u}_1\overline{u}_2 + \overline{v}_1\overline{v}_2)}{\sigma_1\sigma_2} - \frac{2\mu_2(\overline{u}_1\overline{v}_2 - \overline{v}_1\overline{u}_2)}{\sigma_1\sigma_2}\right]\right)$$

$$\times \sum_{k=0}^\infty \epsilon_k \cos\left[k\left(\tan^{-1}\frac{b_1}{a_1} - \tan^{-1}\frac{b_2}{a_2} + \tan^{-1}\frac{\mu_2}{\mu_1}\right)\right] I_k\left(\frac{w_1}{\sigma_1}\sqrt{a_1^2 + b_1^2}\right) I_k\left(\frac{w_2}{\sigma_2}\sqrt{a_2^2 + b_2^2}\right) I_k\left(\frac{\rho}{1-\rho^2}\frac{w_1 w_2}{\sigma_1\sigma_2}\right). \tag{25}$$

Expressed in terms of the magnitude and direction of the mean wind vectors,

$$p(w_1, w_2) = \frac{w_1 w_2}{\sigma_1^2 \sigma_2^2 (1 - \rho^2)} \exp\left(-\frac{1}{2(1-\rho^2)}\left[\frac{w_1^2 + \overline{\mathcal{U}}_1^2}{\sigma_1^2} + \frac{w_2^2 + \overline{\mathcal{U}}_2^2}{\sigma_2^2} - \frac{2\rho\overline{\mathcal{U}}_1\overline{\mathcal{U}}_2\cos(\phi_1 - \phi_2 + \psi)}{\sigma_1\sigma_2}\right]\right)$$

$$\times \sum_{k=0}^\infty \epsilon_k \cos(k\nu) I_k\left(\frac{w_1}{\sigma_1}\sqrt{a_1^2 + b_1^2}\right) I_k\left(\frac{w_2}{\sigma_2}\sqrt{a_2^2 + b_2^2}\right) I_k\left(\frac{\rho}{1-\rho^2}\frac{w_1 w_2}{\sigma_1\sigma_2}\right), \tag{26}$$

where

$$\nu = \tan^{-1}\frac{(1-\rho^2)\dfrac{\overline{\mathcal{U}}_1\overline{\mathcal{U}}_2}{\sigma_1\sigma_2}\sin(\phi_1 - \phi_2 + \psi)}{(1+\rho^2)\dfrac{\overline{\mathcal{U}}_1\overline{\mathcal{U}}_2}{\sigma_1\sigma_2}\cos(\phi_1 - \phi_2 + \psi) - \rho\left(\dfrac{\overline{\mathcal{U}}_1^2}{\sigma_1^2} + \dfrac{\overline{\mathcal{U}}_2^2}{\sigma_2^2}\right)} \tag{27}$$

and

$$\sqrt{a_i^2 + b_i^2} = \frac{1}{1-\rho^2}\left[\frac{\overline{\mathcal{U}}_i^2}{\sigma_i^2} + \rho^2\frac{\overline{\mathcal{U}}_{3-i}^2}{\sigma_{3-i}^2} - \rho\frac{2\overline{\mathcal{U}}_1\overline{\mathcal{U}}_2}{\sigma_1\sigma_2}\cos(\phi_1 - \phi_2 + \psi)\right]^{1/2}. \tag{28}$$





When numerically evaluating the bivariate Rice distribution, it is convenient to transform the infinite series in Eqs. (25) and (26) into an integral which is then approximated numerically (Appendix A).

Note that $p(w_1, w_2)$ depends on the relative orientation of the mean wind vectors and the correlation angle $\psi$ only through the combination $\phi_1 - \phi_2 + \psi$. Because of this symmetry, the quantities $\phi_1 - \phi_2$ and $\psi$ cannot be determined individually
from wind speed data alone. As a result, $p(w_1, w_2)$ is determined by six parameters: $(\overline{\mathcal{U}}_1, \sigma_1, \overline{\mathcal{U}}_2, \sigma_2, \rho, \phi_1 - \phi_2 + \psi)$. For particular applications, it may be appropriate to fix either $\phi_1 - \phi_2$ or $\psi$, allowing the other angle to be estimated from data. For example, when considering the temporal dependence structure of winds assumed to have stationary statistics, it can be assumed that $\phi_1 - \phi_2 = 0$. The bivariate Rice distribution also has the discrete symmetry that it is invariant under the transformation $\phi_1 - \phi_2 + \psi \rightarrow -(\phi_1 - \phi_2 + \psi)$. Note that these symmetries are in addition to the invariance of the distribution of components
(Eq. 5) to the rotation of the coordinate system $\theta_i \rightarrow \theta_i + \Delta\theta$, $i = 1, 2$, under which the angles $\phi_1 - \phi_2$ and $\psi$ are individually invariant.

Integrating over $w_2$ to obtain the marginal distribution for $w_1$ we obtain the univariate Rice distribution

$$\int_0^\infty p(w_1, w_2)\, dw_2 = \frac{w_1}{\sigma_1^2} \exp\left(-\frac{w_1^2 + \overline{\mathcal{U}}_1^2}{2\sigma_1^2}\right) I_0\left(\frac{w_1 \overline{\mathcal{U}}_1}{\sigma_1^2}\right), \tag{29}$$

with mean and variance

$$\mathrm{mean}(w_1) = \sigma_1 \sqrt{\frac{\pi}{2}}\, {}_1F_1\left(-\frac{1}{2}, 1, -\frac{\overline{\mathcal{U}}_1^2}{2\sigma_1^2}\right) \tag{30}$$

$$\mathrm{var}(w_1) = 2\sigma_1^2 + \overline{\mathcal{U}}_1^2 - \sigma_1^2 \frac{\pi}{2}\, {}_1F_1^2\left(-\frac{1}{2}, 1, -\frac{\overline{\mathcal{U}}_1^2}{2\sigma_1^2}\right) \tag{31}$$

(with equivalent expressions for $w_2$ obtained by integrating over $w_1$), where ${}_1F_1(\alpha; \beta; z)$ is the confluent hypergeometric function (Gradshteyn and Ryzhik, 2000). Eq. (29) follows from Eq. (25) using the integral (Gradshteyn and Ryzhik, 2000)

$$\int_0^\infty x e^{-ax^2} I_k(bx) I_k(cx)\, dx = \frac{1}{2a} \exp\left(\frac{b^2 + c^2}{4a}\right) I_k\left(\frac{bc}{2a}\right), \tag{32}$$

Neumann's Theorem (Watson, 1922)

$$\sum_{k=0}^\infty \epsilon_k I_k(x) I_k(y) \cos k\phi = I_0\left(\sqrt{x^2 + y^2 + 2xy\cos\phi}\right), \tag{33}$$

and the fact that

$$\cos\nu = \frac{(1+\rho^2)\dfrac{\overline{\mathcal{U}}_1}{\sigma_1}\dfrac{\overline{\mathcal{U}}_2}{\sigma_2}\cos(\phi_1 - \phi_2 + \psi) - \rho\left(\dfrac{\overline{\mathcal{U}}_1^2}{\sigma_1^2} + \dfrac{\overline{\mathcal{U}}_2^2}{\sigma_2^2}\right)}{(1-\rho^2)^2\sqrt{(a_1^2 + b_1^2)(a_2^2 + b_2^2)}}. \tag{34}$$

Note that each wind speed marginal distribution depends only on the magnitude of the mean wind vector, while the joint
distribution also depends on the angle between the two mean wind vectors. As $\rho \rightarrow 0$ only the first term contributes to the infinite series in Eq. (25), and the joint distribution reduces to the product of the marginals.





The joint moments of the bivariate Rice distribution can be evaluated using the Taylor series expansion:

$$I_k\left(\frac{\rho}{1-\rho^2}\frac{w_1 w_2}{\sigma_1 \sigma_2}\right) = \sum_{j=0}^{\infty} \frac{1}{j!(j+k)!} \left(\frac{\rho}{2(1-\rho^2)}\right)^{2j+k} \left(\frac{w_1}{\sigma_1}\right)^{2j+k} \left(\frac{w_2}{\sigma_2}\right)^{2j+k}, \tag{35}$$

which allows the double integral defining the moments to factorize as the products of individual integrals over $w_1$ and $w_2$ that can be evaluated using

$$\int_0^{\infty} x^{\mu} e^{-\alpha x^2} I_k(\beta x)\, dx = \frac{\beta^k \Gamma\left[(k+\mu+1)/2\right]}{2^{k+1}\alpha^{(\mu+\nu+1)/2}\Gamma(k+1)} {}_1F_1\left(\frac{k+\mu+1}{2}; k+1; \frac{\beta^2}{4\alpha}\right). \tag{36}$$

The resulting expression for the joint moments is

$$\begin{aligned}
\mathsf{E}\{w_1^m w_2^n\} &= 2^{m/2} 2^{n/2} \sigma_1^m \sigma_2^n (1-\rho^2)^{1+m/2+n/2} \exp\left(-\frac{1}{2(1-\rho^2)}\left[\frac{\overline{\mathcal{U}}_1^2}{\sigma_1^2} + \frac{\overline{\mathcal{U}}_2^2}{\sigma_2^2} - \frac{2\rho\overline{\mathcal{U}}_1\overline{\mathcal{U}}_2\cos(\phi_1-\phi_2+\psi)}{\sigma_1\sigma_2}\right]\right) \\
&\times \sum_{l=0}^{\infty}\sum_{k=0}^{l}\left[\epsilon_k \rho^{2l-k}\cos(k\nu)\left((1-\rho^2)\frac{\sqrt{(a_1^2+b_1^2)(a_2^2+b_2^2)}}{2}\right)^k \frac{\Gamma\left(l+\frac{m}{2}+1\right)\Gamma\left(l+\frac{n}{2}+1\right)}{l!(l-k)!(k!)^2}\right. \\
&\times \left. {}_1F_1\left(l+\frac{m}{2}+1; k+1; (1-\rho^2)\frac{(a_1^2+b_1^2)}{2}\right) {}_1F_1\left(l+\frac{n}{2}+1; k+1; (1-\rho^2)\frac{(a_2^2+b_2^2)}{2}\right)\right]
\end{aligned} \tag{37}$$

(Mendes and Yacoub, 2007). When the mean vector winds are equal to zero, only the $k=0$ terms contribute to this expression and Eq. (12) is recovered.

Defining the variables $V_i = \overline{\mathcal{U}}_i/\sqrt{2}\sigma_i$, the correlation coefficient between $w_1$ and $w_2$ is given by

$$\mathrm{corr}(w_1, w_2) = \tag{38}$$

$$\frac{(1-\rho^2)^2 \exp\left(-\frac{V_1^2+V_2^2-2V_1V_2\cos(\phi_1-\phi_2+\psi)}{(1-\rho^2)}\right) G(V_1,V_2,\rho,\phi_1-\phi_2+\psi) - \frac{\pi}{4}{}_1F_1\left(-\frac{1}{2};1;-V_1^2\right){}_1F_1\left(-\frac{1}{2};1;-V_2^2\right)}{\left[1+V_1^2-\frac{\pi}{4}{}_1F_1\left(-\frac{1}{2};1;-V_1^2\right)\right]^{1/2}\left[1+V_2^2-\frac{\pi}{4}{}_1F_1\left(-\frac{1}{2};1;-V_2^2\right)\right]^{1/2}},$$

where

$$\begin{aligned}
G(V_1,V_2,\rho,\phi_1-\phi_2+\psi) &= \sum_{l=0}^{\infty}\sum_{k=0}^{l}\left[\epsilon_k \rho^{2l-k}\cos(k\nu)\left((1-\rho^2)\frac{\sqrt{(a_1^2+b_1^2)(a_2^2+b_2^2)}}{2}\right)^k \frac{\Gamma^2\left(l+\frac{3}{2}\right)}{l!(l-k)!(k!)^2}\right. \\
&\times \left. {}_1F_1\left(l+\frac{3}{2}; k+1; (1-\rho^2)\frac{(a_1^2+b_1^2)}{2}\right){}_1F_1\left(l+\frac{3}{2}; k+1; (1-\rho^2)\frac{(a_2^2+b_2^2)}{2}\right)\right].
\end{aligned} \tag{39}$$

The correlation coefficient $\mathrm{corr}(w_1, w_2)$ depends only on the four quantities $(\overline{\mathcal{U}}_1/\sigma_1, \overline{\mathcal{U}}_2/\sigma_2, \phi_1-\phi_2+\psi, \rho)$. Monahan (2012b) considered the correlation structure of wind speeds using the approximation $\mathrm{corr}(w_1, w_2) \simeq \mathrm{corr}(w_1^2, w_2^2)$. Using the assumed covariance structure of the wind components results in the approximate expression:

$$\mathrm{corr}(w_1, w_2) \simeq \frac{\rho^2 + \left(\frac{\overline{u}_1\overline{u}_2}{\sigma_1\sigma_2} + \frac{\overline{v}_1\overline{v}_2}{\sigma_1\sigma_2}\right)\mu_1 + \left(\frac{\overline{u}_1\overline{v}_2}{\sigma_1\sigma_2} - \frac{\overline{v}_1\overline{u}_2}{\sigma_1\sigma_2}\right)\mu_2}{\sqrt{\left(1+\frac{\overline{u}_1^2+\overline{v}_1^2}{\sigma_1^2}\right)\left(1+\frac{\overline{u}_2^2+\overline{v}_2^2}{\sigma_2^2}\right)}} = \frac{\rho^2 + \frac{\overline{\mathcal{U}}_1\overline{\mathcal{U}}_2}{\sigma_1\sigma_2}\cos(\phi_1-\phi_2+\psi)\rho}{\sqrt{\left(1+\frac{\overline{\mathcal{U}}_1^2}{\sigma_1^2}\right)\left(1+\frac{\overline{\mathcal{U}}_2^2}{\sigma_2^2}\right)}}. \tag{40}$$





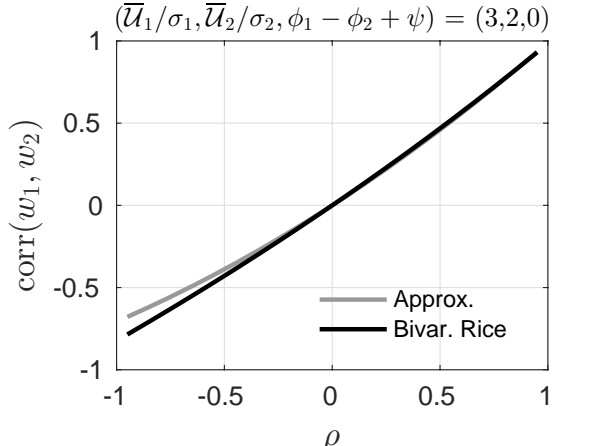 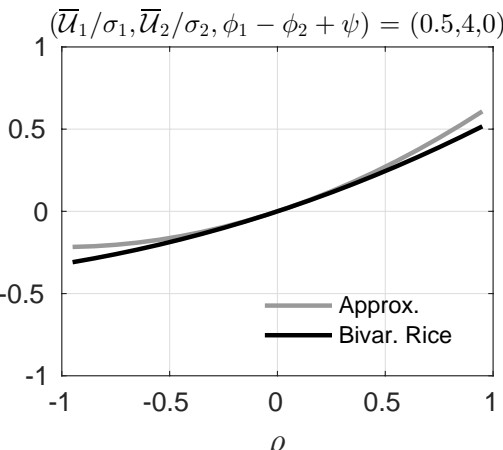

**Figure 2.** Comparison of the correlation coefficient $\mathrm{corr}(w_1, w_2)$ for bivariate Rice distributed variables (Eq. 38) with the approximate expression Eq. (40) for the parameter values $(\overline{\mathcal{U}}_1/\sigma_1, \overline{\mathcal{U}}_2/\sigma_2, \phi_1 - \phi_2 + \psi) = (3,2,0)$ and (0.5,4,0).

Plots of the correlation coefficient (Eq. 38) and the approximation (Eq. 40) as functions of $\rho$ are shown in Figure 2 for $(\overline{\mathcal{U}}_1/\sigma_1, \overline{\mathcal{U}}_2/\sigma_2, \phi_1 - \phi_2 + \psi) = (3,2,0)$ and (1,5,0). Agreement between the exact and approximate values of the correlation coefficient is reasonably good in both cases, with the largest discrepancies generally occurring for larger absolute values of $\rho$.

Examples of the joint Rice pdf (and the associated marginals) are presented in Figure 3 for $(\overline{\mathcal{U}}_1, \sigma_1, \overline{\mathcal{U}}_2, \sigma_2) = (6,4,2,5)$ and $(\rho, \phi_1 - \phi_2 + \psi) = (0.85, \pi), (0,0)$, and (0.85,0). By construction, the marginal distributions are the same in each panel. The distributions of both $w_1$ and $w_2$ are positively skewed, and take respective maxima at values of about $\sigma_1$ and just less than $2\sigma_2$. For the different values of the dependence parameter $\rho$, the joint distributions have considerably different shapes. The joint distribution for $(\rho, \phi_1 - \phi_2 + \psi) = (0.8, \pi)$ is (weakly) negatively skewed, with a nonlinear dependence structure evident in ridges of enhanced probability extending to the left and right upward from the probability maximum. For $(\rho, \phi_1 - \phi_2 + \psi) = (0,0)$, probability contours are concentrated towards smaller values of $w_1$ and $w_2$ (as is the case for the marginal distributions). Finally, $w_1$ and $w_2$ are evidently positively correlated for $(\rho, \phi_1 - \phi_2 + \psi) = (0.8,0)$, with a slight curvature in the shape of the distribution indicating the existence of some nonlinear dependence.

Although the bivariate Rice distribution differs from the bivariate Rayleigh distribution only by allowing for nonzero mean wind vector components, the resulting expressions for the joint pdf (Eq. 25) and the moments (Eq. 37) are much more complicated for the bivariate Rice than the bivariate Rayleigh. Furthermore, while the univariate Rice distribution is a convenient model for the pdf of wind speed, observed winds show clear deviation from Ricean behaviour (e.g. Monahan, 2006, 2007). We will therefore consider another model of the bivariate wind speed distribution with Weibull marginals, which turns out to result in simpler mathematical expressions (at the cost of a more artificial derivation than that of the bivariate Rice distribution).



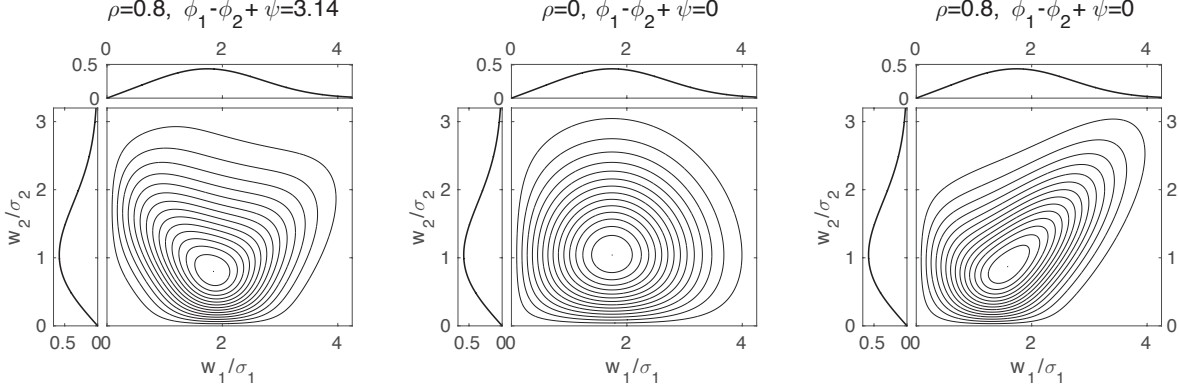

**Figure 3.** As in Figure 1 for the bivariate Rice distribution with $(\overline{\mathcal{U}}_1, \sigma_1, \overline{\mathcal{U}}_2, \sigma_2) = (6, 4, 2, 5)$ and $(\rho, \phi_1 - \phi_2 + \psi) = (0.8, \pi), (0, 0)$, and $(0.8, 0)$.

### 2.3 Bivariate Weibull Distribution

As in Sagias and Karagiannidis (2005) and Yacoub et al. (2005), we obtain the bivariate Weibull distribution from the bivariate Rayleigh distribution through separate power law transformations of $w_1$ and $w_2$. The pdf of a Weibull distributed variable is

$$p(x) = \frac{b}{a}\left(\frac{x}{a}\right)^{b-1}\exp\left(-\left[\frac{x}{a}\right]^b\right), \tag{41}$$

with moments

$$\mathsf{E}\{x^m\} = a^m\Gamma\left(1 + \frac{m}{b}\right), \tag{42}$$

where $a$ and $b$ are denoted the scale and shape parameters, respectively. The Rayleigh distribution is a special case of the Weibull distribution with $a = \sqrt{2}\sigma$ and $b = 2$. Weibull distributed variables remain Weibull under a power-law transformation, with suitably modified scale and shape parameters: if $x$ is Weibull with scale parameter $a$ and shape parameter $b$, $x^k$ will be

be Weibull with scale parameter $a^k$ and shape parameter $b/k$. Thus, a joint wind speed distribution with Weibull marginal distributions can be constructed from a joint Rayleigh distribution using the appropriate power law and scale transformations.

If we start with $(x_1, x_2)$ as bivariate Rayleigh distributed with $\sigma_i = 1/\sqrt{2}$, $i = 1, 2$, we obtain marginal Weibull distributions with specified scale and shape parameters through the transformation

$$w_i = a_i x_i^{2/b_i}. \tag{43}$$

The joint pdfs transform as

$$p(w_1, w_2) = \begin{vmatrix} \partial_{x_1} w_1 & \partial_{x_2} w_1 \\ \partial_{x_1} w_2 & \partial_{x_2} w_2 \end{vmatrix}^{-1} p(x_1, x_2) \tag{44}$$



and so we obtain the bivariate Weibull distribution

$$p(w_1,w_2) = \frac{1}{1-\rho^2}\frac{b_1 b_2}{a_1 a_2}\left(\frac{w_1}{a_1}\right)^{b_1-1}\left(\frac{w_2}{a_2}\right)^{b_2-1}\exp\left(-\frac{1}{(1-\rho^2)}\left[\left(\frac{w_1}{a_1}\right)^{b_1}+\left(\frac{w_2}{a_2}\right)^{b_2}\right]\right)I_0\left(\frac{2\rho}{(1-\rho^2)}\left(\frac{w_1}{a_1}\right)^{b_1/2}\left(\frac{w_2}{a_2}\right)^{b_2/2}\right).$$

(45)

An analagous approach to constructing bivariate Weibull distributions through nonlinear transformations of a bivariate Gaussian was followed in Villanueva et al. (2013); the resulting expressions are considerably more complicated than those considered here.

Evidently, $p(w_1,w_2)$ factorizes into the product of the marginal distributions as $\rho \to 0$. As $\rho \to 1$, we can use Eq. (10) to make the approximation

$$\begin{aligned}p(w_1,w_2) &\simeq \frac{1}{1-\rho^2}\frac{b_1 b_2}{a_1 a_2}\left(\frac{w_1}{a_1}\right)^{b_1-1}\left(\frac{w_2}{a_2}\right)^{b_2-1}\exp\left(-\frac{1}{(1-\rho^2)}\left[\left(\frac{w_1}{a_1}\right)^{b_1}+\left(\frac{w_2}{a_2}\right)^{b_2}\right]\right)\\ &\quad\times\left(\frac{4\pi\rho}{(1-\rho^2)}\left(\frac{w_1}{a_1}\right)^{b_1/2}\left(\frac{w_2}{a_2}\right)^{b_2/2}\right)^{-1/2}\exp\left(\frac{2\rho}{(1-\rho^2)}\left(\frac{w_1}{a_1}\right)^{b_1/2}\left(\frac{w_2}{a_2}\right)^{b_2/2}\right)\\ &\simeq\left[\frac{b_1}{a_1}\left(\frac{w_1}{a_1}\right)^{b_1-1}\exp\left(-\left(\frac{w_1}{a_1}\right)^{b_1}\right)\right]\left[\frac{b_2}{2a_2}\left(\frac{w_1}{a_1}\right)^{-b_1/4}\left(\frac{w_2}{a_2}\right)^{3b_2/4-1}\right]\delta\left(\left(\frac{w_1}{a_1}\right)^{b_1/2}-\left(\frac{w_2}{a_2}\right)^{b_2/2}\right)\\ &=p(w_1)\delta\left(w_2-a_2\left(\frac{w_1}{a_1}\right)^{b_1/b_2}\right),\end{aligned}$$

(46)

where the last equality follows from the fact that

$$\delta\left(f(x)-\alpha\right)=\frac{1}{f'(\alpha)}\delta\left(x-f^{-1}(\alpha)\right).$$

(47)

As expected, $w_1$ and $w_2$ are completely dependent in the limit that $\rho \to 1$ (although they are not perfectly correlated if $b_1 \neq b_2$ as the functional relationship

$$w_2 = a_2\left(\frac{w_1}{a_1}\right)^{b_1/b_2}$$

(48)

between the two variables will be nonlinear).

The relatively simple form of the bivariate Weibull distribution permits a relatively simple expression for the conditional distribution

$$\begin{aligned}p(w_2|w_1) &= \frac{p(w_1,w_2)}{p(w_1)}\\ &=\left\{\frac{b_2}{a_2}\left(\frac{w_2}{a_2}\right)^{b_2-1}\exp\left(-\left[\frac{w_2}{a_2}\right]^{b_2}\right)\right\}\\ &\quad\times\left\{\frac{1}{1-\rho^2}\exp\left(-\frac{\rho^2}{1-\rho^2}\left[\left(\frac{w_1}{a_1}\right)^{b_1}+\left(\frac{w_2}{a_2}\right)^{b_2}\right]\right)I_0\left(\frac{2\rho}{(1-\rho^2)}\left(\frac{w_1}{a_1}\right)^{b_1/2}\left(\frac{w_2}{a_2}\right)^{b_2/2}\right)\right\}.\end{aligned}$$

(49)




The factor in the second set of braces characterizes how conditioning on the value of $w_1$ changes the distribution of $w_2$ from its marginal distribution (and corresponds to a copula density; e.g. Schlözel and Friederichs, 2008). Note that for $w_1$ sufficiently large we can write

$$p(w_2|w_1) \simeq \frac{1}{\sqrt{4\pi\rho(1-\rho^2)}} \frac{b_2}{a_2} \left(\frac{w_2}{a_2}\right)^{3b_2/4-1} \left(\frac{w_1}{a_1}\right)^{-b_1/4} \exp\left(-\frac{1}{1-\rho^2}\left[\rho\left(\frac{w_1}{a_1}\right)^{b_1/2} - \left(\frac{w_2}{a_2}\right)^{b_2/2}\right]^2\right). \tag{50}$$

For $\rho$ not too close to zero, the conditional distribution for large $w_1$ is concentrated around the nonlinear regression curve

$$\left(\frac{w_2}{a_2}\right)^{b_2} = \rho^2 \left(\frac{w_1}{a_1}\right)^{b_1}. \tag{51}$$

Computing the moments, we obtain

$$\mathsf{E}\{w_1^m w_2^n\} = a_1^m a_2^n \Gamma\left(1+\frac{m}{b_1}\right)\Gamma\left(1+\frac{n}{b_2}\right) {}_2F_1\left(-\frac{m}{b_1}, -\frac{n}{b_2}, 1; \rho^2\right). \tag{52}$$

(Sagias and Karagiannidis, 2005; Yacoub et al., 2005). The correlation coefficient is then given by

$$\mathrm{corr}(w_1, w_2) = \frac{\Gamma\left(1+\frac{1}{b_1}\right)\Gamma\left(1+\frac{1}{b_2}\right)\left[{}_2F_1\left(-\frac{1}{b_1}, -\frac{1}{b_2}, 1; \rho^2\right) - 1\right]}{\sqrt{\left[\Gamma\left(1+\frac{2}{b_1}\right) - \Gamma^2\left(1+\frac{1}{b_1}\right)\right]\left[\Gamma\left(1+\frac{2}{b_2}\right) - \Gamma^2\left(1+\frac{1}{b_2}\right)\right]}}. \tag{53}$$

For $b_1, b_2 > 1$ (the relevant range of shape parameters for wind speeds), ${}_2F_1(-1/b_1, -1/b_2, 1; \rho^2)$ is an increasing function of $\rho^2$ with ${}_2F_1(-1/b_1, -1/b_2, 1; 0) = 1$. Therefore, the bivariate Weibull distribution is unable to represent situations in which the wind speeds are negatively correlated.

Examples of the bivariate Weibull distribution for $(a_1, b_1, a_2, b_2) = (4, 1.5, 5, 7)$ and values of $\rho = 0, 0.7$, and $0.95$ are shown in Figure 4. Again, the marginal distributions in all three cases are the same by construction. The distribution of $w_1$ is positively skewed with a maximum near a value of $w_1 = 0.5a_1$, while that of $w_2$ is negatively skewed with a maximum near $w_2 = a_2$. For $\rho = 0$ the joint distribution is simply the product of the marginals. As $\rho$ increases, $w_1$ and $w_2$ become positively correlated - although the correlation is weak even for $\rho = 0.7$ for this set of parameter values. At the value of $\rho = 0.95$, while the correlation of the two variables is only moderate, a strong nonlinear dependence is evident in the concentration of the distribution around the curve given by Eq. (48).

## 3 Fits of bivariate Rice and Weibull distributions to observed wind speeds

Many wind datasets from different locations are available, and it is impracticable to consider joint distributions of wind speeds from all of these. In this Section, we will consider examples of the joint distribution of wind speeds using data from a representative range of settings. Bivariate distributions of wind speeds at both different locations in space and different points in time will be considered. The sampling of the wind speeds considered will be temporal (that is, individual samples will correspond to a specific time for spatial joint pdfs and a specific pair of times for temporal joint pdfs). Best-fit values of the parameters of the bivariate Weibull and Rice distributions we present (Table 1) were obtained numerically as maximum likelihood estimates, with $\phi_1 - \phi_2$ set to zero. Goodness-of-fit of the distributions was assessed using a statistical test described in Appendix B.



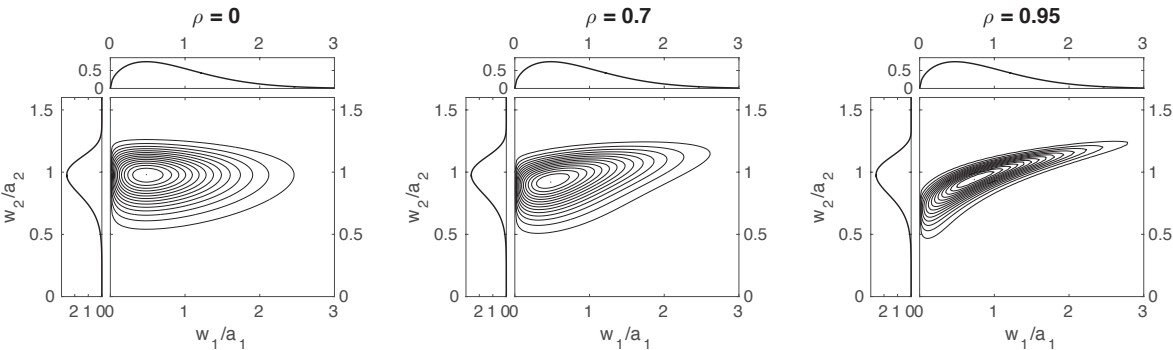

**Figure 4.** As in Figure 1 for the bivariate Weibull distribution with $(a_1, b_1, a_2, b_2) = (4, 1.5, 5, 7)$ and $\rho = 0, 0.7$, and $0.95$ and the speeds $w_1, w_2$ scaled respectively by the Weibull scale parameters $a_1, a_2$.

### 3.1 Wind speeds at 500 hPa

We first consider the joint distribution of 00Z December, January, and February 500 hPa wind speeds from 1979 to 2014. These data were taken from the European Centre for Medium-Range Weather Forecasts (ECMWF) ERA-Interim Reanalysis (Dee et al., 2011), subsampled to every second day to minimize the effect of serial dependence on the goodness-of-fit test

(Monahan, 2012b). The wind speed data were computed as the magnitude of zonal and meridional components.

The joint distributions of wind speeds at four pairs of latitudes along 216°W are presented in Figure 5. A moderately strong negative correlation ($r = -0.55$) is evident between wind speeds at (39°S,216°W) and (54.75°S,216°W) (Figure 5a). Because it is unable to model a negative correlation between wind speeds, the best-fit bivariate Weibull pdf differs substantially from the distribution of the observed winds (Figure 5e). The goodness-of-fit test correspondingly rejects the null hypothesis that the

observations are drawn from this distribution ($p = 0$). In contrast, the bivariate Rice distribution provides a reasonable model of the joint distribution of wind speeds at these two locations (Figure 5i) and the goodness-of-fit test provides no evidence that these data are statistically incompatible with this distribution ($p = 0.31$). For wind speeds at these two locations, the bivariate Rice distribution is evidently a better model than the bivariate Weibull distribution.

In contrast, for wind speeds at (12°N, 216°W) and (15.75°N, 216°W), the null hypotheses of being drawn from either the

bivariate Rice or Weibull distributions are rejected at the 95% significance level for both distributions. These wind speeds are weakly correlated ($r = 0.37$) but show evidence of nonlinear dependence. The joint pdf of the observed speeds is characterized by two ridges of high probability extending to the right, upward and downward away from the region of maximum probability (Figure 5b,f,j). These ridges are not captured by either of the best-fit bivariate Weibull or Rice distributions, although a hint of this structure is evident in the Rice distribution. While the fit of one or both of these parametric distributions to these observed

wind speed data may be sufficiently good for practical applications, nevertheless we can confidently exclude the possibility that these data are drawn from either distribution.



| | | Sample Size | Bivariate Rice $(\overline{\mathcal{U}}_1, \sigma_1, \overline{\mathcal{U}}_2, \sigma_2, \phi_1 - \phi_2 + \psi, \rho)$ ( $\mathrm{ms}^{-1}$ , $\mathrm{ms}^{-1}$ , $\mathrm{ms}^{-1}$ , $\mathrm{ms}^{-1}$ , - , - ) | Bivariate Weibull $(a_1, b_1, a_2, b_2, \rho)$ ( $\mathrm{ms}^{-1}$ , - , $\mathrm{ms}^{-1}$ , - , - ) |
|---|---|---|---|---|
| 500 hPa | (39°S,54.75°S) | 1625 | (19.8, 10.5, 17.9, 11.3, 3.14, 0.71) | (25.6, 2.5, 24.5, 2.3, 0.0) |
| | (12°N,15.75°N) | 1625 | (0.0, 4.8, 6.9, 6.2, 0.88, 0.79) | (6.8, 1.9, 11.0, 2.0, 0.62) |
| | (3°S, 0.75°N) | 1625 | (3.8, 3.1, 5.5, 3.5, 0.0, 0.75) | (5.8, 2.1, 7.6, 2.3, 0.85) |
| | (15°S, 45°S) | 1625 | (1.1, 4.2, 26.9, 11.0, 0.0, 0.15) | (6.0, 1.8, 32.7, 3.1, 0.23) |
| 500 hPa | 0.5 day lag | 1625 | (19.4, 10.8, 19.4, 10.5, 0, 0.80) | (25.1, 2.2, 24.9, 2.3, 0.90) |
| | 1.5 day lag | 1624 | (19.4, 10.9, 19.7, 10.4, 0, 0.40) | (25.5, 2.4, 25.4, 2.5, 0.62) |
| | 3 day lag | 1623 | (19,3, 10.9, 19.4, 10.4, 0, 0.23) | (25.6, 2.5, 25.3, 2.6, 0.44) |
| Cabauw | night | 1189 | (1.9, 1.9, 6.9, 3.7, 0.0, 0.85) | (3.1, 1.7, 8.9, 2.3, 0.92) |
| | night $R_1$ | 763 | (1.6, 1.2, 5.0, 4.0, 0.0, 0.83) | (2.3, 2.2, 7.6, 2.1, 0.90) |
| | night $R_2$ | 427 | (3.5, 2.0, 9.4, 3.0, 0.0, 0.90) | (4.6, 2.2, 10.9, 3.6, 0.95) |
| | day | 1060 | (3.2, 2.8, 3.6, 4.7, 0.12, 0.98) | (5.0, 2.3, 7.3, 2.1, 0.99) |
| QuikSCAT | (6.5°S,162°W) | 321 | (7.2,1.5,6.7,1.7,1.27,0.48) | (8.0,5.4,7.6,4.8,0.35) |
| | (6.5°S,152°W) | 185 | (7.3, 1.5, 7.0,1.7,1.15,0.95) | (8.0,5.6,7.8,4.9,0.66) |
| | (6.5°S,142°W) | 208 | (7.1,1.6,7.2,1.6,0.60,0.80) | (7.9,5.1,8.0,5.0,0.83) |

**Table 1.** Maximum likelihood parameter estimates for the wind speed data shown in Figures 5, 6, 7, 8.

The wind speeds at (3°S, 216°W) and (0.75°N, 216°W) are correlated ($r = 0.68$) and their scatter clusters around a straight line extending away from the origin (Figure 5c). Both the best-fit bivariate Weibull and Rice distributions appear to the eye to be good fits to the data (Figures 5g,k), and in neither case can the null hypothesis be rejected that the data are drawn from these distributions. Only small differences exist between the two best-fit distributions for these data.





**Figure 5.** Joint distributions of 500 hPa DJF 00Z wind speeds at four different pairs of latitudes along $216°$W. Upper row: scatterplots of wind speed data. Middle row: maximum likelihood bivariate Weibull pdfs (white contours) and kernel density estimates of the observed joint pdf (colours). The $p$ value of a goodness-of-fit test with the null hypothesis that the observed wind speed data are drawn from the corresponding best-fit bivariate Weibull distribution is given. Bottom row: as in the middle row, for the best-fit bivariate Rice distribution. Values of the best-fit model parameters are given in Table 1.

Finally, the wind speeds at $(15°S, 216°W)$ and $(45°S, 216°W)$ are uncorrelated $(r = 0.06)$ and fit sufficiently well by both the bivariate Weibull and Rice distributions (Figure 5d,h,l) that in neither case is the null hypothesis rejected. As in the previous example, the bivariate best-fit Rice and Weibull distributions are essentially indistinguishable for these data.





Considering the spatial correlation structure of these 500 hPa winds, we find cases in which one distribution (the Rice) is evidently a better fit to the data than the other (the Weibull); in which neither distribution provides a statistically significant fit to the data; and in which both distributions fit the data equally well.

The temporal dependence structure of the wind speed at ($39°$S,$216°$W) is illustrated in Figure 6. As with the previous
calculations, the pairs of lagged wind speeds $(w_n, w_{n+s})$ were subsampled to two-day resolution to minimize the effect of serial dependence on the results of the goodness-of-fit tests. As the lag increases, the value of the dependence parameter $\rho$ decreases as expected for both the Weibull and Rice distributions (Figure 6h,l). For most lags, the null hypothesis of the Weibull distribution as a model for the joint distribution is rejected (as $p < 0.05$; Figure 6d). The rejection of the null hypothesis of a bivariate Weibull distribution is most robust for lags shorter than 3 days. In contrast, the null hypothesis of a bivariate Rice
distribution is rejected less often than it is not - although $p < 0.05$ for more than one third of the lags (Figure 6d). Inspection of the example distributions shown demonstrates that the bivariate Weibull distributions are broader around their principal axis for small to intermediate wind speeds in a way that is not consistent with observations (Figure 6e-g). Such structures are not seen in the best-fit Rice distributions (Figure 6i-k). Note that for lags of 0.5 and 1.5 days the observed distributions suggest a flaring out of the joint distribution for large wind speeds that is accounted for by neither the bivariate Weibull nor Rice distributions.
There is good evidence that these data were not drawn from a bivariate Weibull distribution, and the evidence that they are drawn from a bivariate Rice distribution is not strong.

## 3.2  Wind speeds over land at 10 m and 200 m

Wind speeds at altitudes of $10$ m and $200$ m measured from a $213$ m tower in Cabauw, Netherlands ($51.971°$N,$4.927°$E) main-tained by the Cabauw Experimental Site for Atmospheric Research (CESAR;  van Ulden and Wieringa, 1996) are available
with 10-minute resolution from 1 January 2001 through 31 December 2012. We will focus on data from July, August, and September (JAS), separated into daytime (08:00-16:00 UTC) and nighttime (20:00-05:00) periods. These data were subsam-pled in time to account for serial dependence. Only every 50th point was used in the following analysis. A small number of zero wind speed values were removed from the dataset.

Monahan et al. (2011, 2015) demonstrated the existence of two distinct regimes of the nocturnal boundary layer in these data,
corresponding to the very and weakly stratified boundary layers (vSBL and wSBL; e.g. Mahrt, 2014). These regimes, denoted respectively $R_1$ and $R_2$ were separated in Monahan et al. (2015) using a two-state Hidden Markov Model (HMM). Conditioning the data on the HMM state, the scatterplot of wind speeds at 10m and 200m separates into two distinct populations (Figure 7a-c). The structure of the boundary layer shows no evident regime structure during the day (Figure 7d). In all cases, the wind speeds at the two altitudes are highly correlated.
Maximum likelihood estimates of the bivariate Weibull and Rice distributions for the full nighttime data show evident dis-agreement between the scatter of the data and the best-fit distributions (Figure 7e,i). Goodness-of-fit tests for both distributions were rejected (with $p = 0$ in both cases). When conditioned on being in either regime $R_1$ or $R_2$, both the bivariate Rice and Weibull distributions result in much better representations of the data (Figure 7f,g,j,k). In all cases the $p$ values exceed 0.05, so the fits cannot be rejected at the 95% significance level. While neither the bivariate Rice nor Weibull distributions is a good





**Figure 6.** The temporal dependence structure of 500 hPa DJF 00Z wind speeds at (39°S,216°W) for lags from 0.25 to 4 days. Upper row (a)-(c): scatterplots of wind speeds separated by 0.5 day (first column), 1.5 days (second column), and 3 days (third column). Middle row (e)-(g): maximum likelihood bivariate Weibull distribution (white contours) and kernel density estimate of the joint pdf of the lagged data. The $p$-value of a goodness-of-fit test of the bivariate Weibull fit is quoted in white. Bottom row (i)-(j) as in the middle row (e)-(g) for the bivariate Rice distribution. Panels (d,h,l) show results at a range of different time lags. Panel (d): $p$-values of bivariate goodness-of-fit tests for bivariate Weibull (black) and bivariate Rice (red). The thin black curve is the 0.95 significance level. Panel (h): best-fit estimate of the parameter $\rho$ of the bivariate Weibull distribution. Panel (l): best-fit estimates of $\rho$ (black) and $\cos\phi$ (blue) for the Rice distribution. Values of the best-fit model parameters are given in Table 1.

probability model for the joint distribution of wind speeds at these two altitudes, both are reasonable models for the distributions conditioned by regime occupation. Finally, the fits of neither the bivariate Rice and Weibull distributions to the daytime

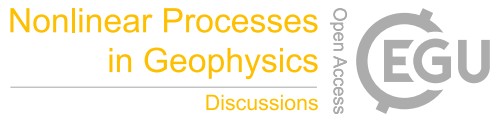

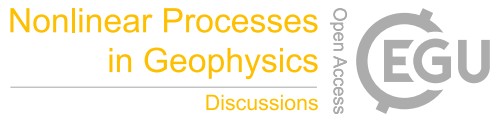

**Figure 7.** Joint distribution of JAS wind speeds at 10m and 200m measured at Cabauw, NL. Upper row: Wind speed scatterplots. Middle row: kernel density estimate of the joint pdf of wind speeds (colour) and maximum likelihood bivariate Weibull distribution (contours). The $p$-value of a goodness of fit test of the bivariate Weibull distribution is quoted in white. Bottom row: as in middle row, but for best-fit bivariate Rice distribution (contours). First column: all nighttime data (20:00-05:00 UTC). Second column: nighttime data conditioned on being in regime $R_2$ (very stable boundary layer). Third column: nighttime data conditioned on being in regime $R_1$ (weakly stable boundary layer). Fourth column: all daytime data (08:00-16:00). Values of the best-fit model parameters are given in Table 1.

data are statistically significant at the 95% significance level. In particular, the joint Rice distribution is too broad (relative to observations) for small values of $w_{10}$ and $w_{200}$ and neither distribution is broad enough at higher wind speed values.


### 3.3 Sea surface wind speeds

Twice-daily December, January, and February level 3.0 gridded SeaWinds scatterometer equivalent neutral 10 m wind speeds between 60°S and 60°N at a resolution of $0.25° \times 0.25°$ from the National Aeronautics and Space Administration (NASA) Quick Scatterometer (QuickSCAT; Perry, 2001) are available from December 1999 through February 2008. Data flagged as
having possibly been corrupted by rain were excluded from the following analysis. Although the data are nominally twice-daily, it is often the case that data for either the ascending or the descending pass of the satellite are missing. The maximum likelihood parameter estimates of bivariate wind speed distributions and goodness-of-fit tests were carried out using every third non-missing data point in order to minimize the effect of serial dependence. For the goodness-of-fit tests $M = 10$ quantiles were used because of the relatively small sample sizes. Because of the near-polar orbit of the satellite results, observations of
wind speed at different locations are not simultaneous. The joint distributions we consider therefore combine dependence in both space and time.

Joint distributions of wind speed at (6.5°S,135°W) with speeds at points along a zonal transect to 165°W (in increments of 1°) were estimated. As the distance between the two positions increases, there is a decreasing trend in the best-fit bivariate Weibull dependence parameter $\rho$ (Figure 8h) with some small fluctuations likely due to sampling variability. The same is not
true for the best-fit bivariate Rice dependence parameter $\rho$, which fluctuates wildly (Figure 8l). Large fluctuations also seen in the best-fit value of $\cos\psi$ are clearly correlated with those of $\rho$: where one parameter is anomalously large (relative to the spatial trend), the other is anomalously small. Of the 30 pairs of points considered, the null hypothesis of a bivariate Rice distribution is rejected (at the 95% significance level) at only one (Figure 8d). In contrast, $p < 0.05$ for the bivariate Weibull distribution at several longitudes, particularly close to the base point at (6.5°S,135°W). Inspection of the best-fit bivariate
Weibull pdfs (Figure 8e-g) shows that they are broader for smaller wind speeds than for larger values, a feature not evident in the best-fit bivariate Rice distributions (Figure 8i-k) or the scatter of data (Figure 8a-c). From these results, we find only equivocal evidence that the pairs of wind speed data along this zonal transect are drawn from a bivariate Weibull distribution and no strong evidence to reject the null hypothesis that they are drawn from a bivariate Rice distribution. In fact, the surface wind vector components in the tropics are known to be non-Gaussian (e.g. Monahan, 2007) so we have a priori reasons to
believe the joint distribution should not be Ricean. The fact that the data do not allow for a rejection of the null hypothesis that the winds are bivariate Rice is likely a consequence of the relatively small sample size.

The large variations in best-fit estimates of $\rho$ and $\cos\psi$ for the bivariate Rice distribution result from the fact that for some parameter values the distribution is only weakly sensitive to simultaneous changes in these parameters: increases in $\rho$ can be offset by decreases in $\cos\phi$ with only small changes to the joint distribution. To demonstrate this weak sensitivity, 50
realizations of bivariate Ricean variables with $(\overline{\mathcal{U}}_1, \sigma_1, \overline{\mathcal{U}}_2, \sigma_2, \rho, \phi_1 - \phi_2 + \psi) = (7.3, 1.5, 6.9, 1.5, 0.5, 0)$ were generated for each of the sample sizes of $N = 250, 1500,$ and $9000$. Maximum likelihood estimates of these parameters obtained from these realizations demonstrate that for the smaller samples $\rho$ and $\cos\psi$ show strong and correlated sampling variability, with large increases in $\rho$ combined with large decreases in $\cos\psi$ (Figure 9a-c). As expected, these sampling fluctuations become smaller

**Figure 8.** Joint distribution of DJF QuikSCAT wind speeds at (6.5°S, 135°W) and westward along a zonal transect to 165°W. Upper row (a)-(c): Scatterplots of wind speed at (6.5°S, 135°W) and at the points (6.5°S,162°W), (6.5°S,152°W),(6.5°S,142°W). Middle row (e)-(g): Kernel density estimate of the joint pdf (colour) as well as the maximum likelihood bivariate Weibull distribution (white contours). The $p$-value of a goodness of fit test of the bivariate Weibull distribution is quoted in white. Bottom row (f)-(k): As in (e)-(g) but for the bivariate Rice distribution. Panel (d): $p$-values of the bivariate goodness-of-fit tests for the wind speeds along the transect, for the bivariate Weibull distribution (black) and the bivariate Rice distribution (red). Panel (h): estimate of the parameter $\rho$ from the best-fit bivariate Weibull distribution along the transect. Panel (l): estimate of the parameters $\rho$ (black) and $\cos\psi$ (blue) from the best-fit bivariate Rice distribution along the transect. Values of the best-fit model parameters are given in Table 1.



**Figure 9.** Upper row: Estimates of $\rho$ and $\cos\psi$ from 50 realizations of bivariate Ricean variables with $(\overline{\mathcal{U}}_1, \sigma_1, \overline{\mathcal{U}}_2, \sigma_2, \rho, \phi_1 - \phi_2 + \psi) =$ $(7.3, 1.5, 6.9, 1.5, 0.4, 0)$ for each of the sample sizes $N = 250, 1250$ and $9000$. The open circles correspond to estimates with $\cos(\phi_1 - \phi_2 + \psi) \geq 0.99$, while the stars are estimates with $\cos(\phi_1 - \phi_2 + \psi) < 0.99$. Middle row: contours of bivariate Rice pdfs corresponding to 10 of the 50 best-fit parameter estimates (randomly chosen). The contour values are the same for all pdfs within each subplot. Bottom row: scatterplot of the sample correlation coefficient between the two Ricean variables and the correlation coefficient given by the approximate expression Eq. (40). The 1:1 line is given in solid black. The open circles and stars are as in the upper row.

as the sample size increases. Despite the large variation of $\rho$ and $\cos\psi$ for small to intermediate sample sizes, there is relatively little variation in the structure of the corresponding bivariate Rice distributions (Figure 9d-f).

An indication of why increases in $\rho$ should counterbalance decreases in $\cos\psi$ with only small effects on the joint distribution is given by the approximate expression for the correlation coefficient, Eq. (40). The value of this approximation is unaffected



by changes in $\rho$ and $\psi$ that leave the numerator invariant. The compensation between sampling variations in $\rho$ and $\cos\psi$ is evident the fact that $\text{corr}(w_1, w_2)$ given by Eq. (40) is an excellent approximation to the sample correlation coefficient even for estimates of $\rho$ and $\cos\phi$ which are far away from the population values (Figure 9g-i). Note that there is no evident relationship between sampling fluctuations in $\text{corr}(w_1, w_2)$ and those of $\rho$ and $\psi$: the range of sample correlation values for $(\rho, \cos\psi)$ near

the population values of $(0.5, 1)$ (open circles) is the same as that for values of $(\rho, \cos\psi)$ far from these values (stars). In these parameter ranges, the dependence between $w_1$ and $w_2$ constrains $\rho$ and $\psi$ not individually, but together - over large ranges of values for sufficiently small sample sizes.

## 4   Conclusions

This study has considered two idealized probability models for the joint distribution of wind speeds, both derived from models

for the joint distribution of the horizontal wind components. The first, the bivariate Rice distribution, follows from assuming that the wind vector components are bivariate Gaussian with an idealized covariance structure. The second, the bivariate Weibull distribution, arises from nonlinear transformations of variables with a bivariate Rice distribution in the limit that the mean vector winds vanish (the bivariate Rayleigh distribution). While the bivariate Rice distribution has the advantage of being more flexible and naturally related to a simplified model for the joint distribution of the wind components, the bivariate Weibull

distribution is mathematically much simpler and easier to work with. Through consideration of a range of joint distributions of observed wind speeds (over land and over the ocean; at the surface and aloft; in space and in time) the bivariate Rice distribution was shown to generally model the observations better than the bivariate Weibull distribution. However, in many circumstances the differences between the two distributions are small and the convenience of the bivariate Weibull distribution relative to the bivariate Rice distribution is a factor which may motivate its use.

Many of the assumptions that have been made regarding the distribution of the wind components are known not to hold in various settings. For example, the vector wind components are generally not Gaussian, either aloft or at the surface (e.g. Monahan, 2007; Luxford and Woollings, 2012; Perron and Sura, 2013), and fluctuations will not generally be isotropic (especially over land, cf. Mao and Monahan, 2017). Furthermore, when used to model temporal dependence the assumed correlation structure cannot account for the anisotropy in autocorrelation of orthogonal components in either space (e.g. Buell, 1960) or

time (e.g Monahan, 2012b). Relaxing the assumptions regarding isotropy of correlation structure results in expressions for the joint speed distributions involving integrals over angle which are not analytically tractable.

While it is possible to relax the assumption of Gaussian components for univariate speed distribution (e.g. Monahan, 2007; Drobinski et al., 2015), extending this analysis to the bivariate case would involve specifying a non-Gaussian dependence structure for the components. At present, there is no physically-based model for such dependence. Without any such physi-

cal justification, the only option is an empirical investigation of the ability of different copula models to represent observed joint wind speed distributions. It is unlikely that a copula-based model for dependence of components will admit analytically tractable expressions for joint speed distributions. A copula-based analysis on either the components or the speeds directly is also likely necessary for modelling extreme wind speeds (either large percentiles, peaks over threshold, or block maxima), as



the tails of the bivariate Rice and Weibull distributions may not be adequate for this task. Finally, extending the explicit closed-form results for the bivariate wind speed distribution to a higher-dimensional multivariate setting - of wind speeds alone, or of a mixture of wind speeds and other meteorological quantities - will be analytically intractable for any except the simplest (and likely unrealistic) covariance structures. It may not be practical to extend the program of obtaining closed-form expressions for

joint speed distributions from models of the component distributions beyond the bivariate Rice and Weibull speed distributions considered in this study.

Ultimately, it would be best for models of the joint distribution of wind speeds to arise from physically-based (if still idealized) models, as has been done for the univariate case in Monahan (2006) and Monahan et al. (2011). The development of such models represents an interesting direction of future study.

*Data availability.* The ERA-Interim 500 hPa zonal and meridional wind components were obtained from
`http://www.ecmwf.int/en/research/climate-reanalysis/era-interim`). The Cabauw tower data were downloaded
from `http://www.cesar-database.nl/`. The Level 3.0 QuickSCAT data were downloaded from the NASA Jet Propulsion Laboratory Physical Oceanography Distributed Active Archive Center, `http://podaac.jpl.nasa.gov/dataset/QSCAT_LEVEL_3_V2`.

### Appendix A: Numerical computation of bivariate Rice pdf

Equation (25) is difficult to evaluate numerically when the arguments of the Bessel functions become large. We have found that a computationally more stable result is obtained when this equation is expressed in the form

$$p(w_1, w_2) = \frac{w_1 w_2}{2\pi \sigma_1^2 \sigma_2^2 (1-\rho^2)} \int\limits_0^{2\pi} \exp f(w_1, w_2, \lambda) \, d\lambda, \tag{A1}$$

where

$$f(w_1, w_2, \phi) = -\frac{1}{2(1-\rho^2)} \left[ \frac{w_1^2 + \overline{\mathcal{U}}_1^2}{\sigma_1^2} + \frac{w_2^2 + \overline{\mathcal{U}}_2^2}{\sigma_2^2} - \frac{2\rho \overline{\mathcal{U}}_1 \overline{\mathcal{U}}_2 \cos(\phi_1 - \phi_2 + \psi)}{\sigma_1 \sigma_2} \right] + \ln I_0 \left( \sqrt{A^2 + B^2 + 2AB \cos\lambda} \right) \tag{A2}$$

$+ C\cos\nu \cos\lambda + \ln\cosh(C\sin\nu \sin\lambda) \, d\lambda,$

with

$$A = \frac{w_1 \sqrt{a_1^2 + b_1^2}}{\sigma_1} \tag{A3}$$

$$B = \frac{w_2 \sqrt{a_2^2 + b_2^2}}{\sigma_2} \tag{A4}$$

$$C = \frac{\rho}{1-\rho^2} \frac{w_1 w_2}{\sigma_1 \sigma_2}, \tag{A5}$$





and the integral is evaluated numerically. Eq. (A1) is obtained from Eq. (26) using the fact that

$$
I_k(x)I_k(y) = \frac{1}{2\pi} \int\limits_0^{2\pi} I_0(\sqrt{x^2 + y^2 + 2xy\cos\phi})\cos k\phi \; d\phi
\tag{A6}
$$

(which follows from Eq. 33) and use of Eq. (24). When $w_i/\sigma_i$ becomes large, numerical evaluations of the Bessel function in Eq. (A1) become unreliable. For the present computations using Matlab, values of Inf occur in such cases. This problem was not solved by using the approximation Eq. (10) when the argument of the Bessel function is large, as this approximation is not sufficiently accurate.

## Appendix B: Bivariate goodness-of-fit test

Goodness-of-fit of the bivariate distributions considered was assessed as follows. For the speed dataset $w_{j,n}$, $j = 1,2$ and $n = 1,...,N$, evenly spaced quantiles $q_{j,i} = i/M$, $i = 0,...,M$ for the marginals are estimated. The quantiles $q_{j,0} = 0$ and $q_{j,M} = 1$ are estimated respectively as 0.9 times the smallest observed value and 1.1 times the largest observed value. The number of pairs of observations falling simultaneously into all pairs of quantiles are computed:

$$
f_{kl} = \sum_{n=1}^N \mathbf{1}\big[\big(w_{1,n} \in (q_{1,k}, q_{1,(k+1)}]\big) \cap \big(w_{2,n} \in (q_{2,l}, q_{2,(l+1)}]\big)\big],
\tag{B1}
$$

where $\mathbf{1}(\cdot)$ is the indicator function. The pdf with maximum likelihood parameters $\theta$ is then integrated to obtain the expected number of observations in these intervals

$$
g_{kl} = N \int\limits_{q_{1,k}}^{q_{1,(k+1)}} \int\limits_{q_{2,l}}^{q_{2,(l+1)}} p(w_1, w_2; \theta) \; dw_1 dw_2
\tag{B2}
$$

and the test statistic $A$ is computed:

$$
A = \frac{1}{M^2} \sum_{k,l=1}^M |f_{kl} - g_{kl}|.
\tag{B3}
$$

Any elements $g_{kl}$ which take the value of Inf (because of numerical difficulties in evaluating the Bessel function for large arguments; cf. Appendix A) are excluded from the calculation of the test statistic $A$.

After computation of $A$ from the observations, an ensemble of $B$ random realizations of length $N$ from $p(w_1, w_2; \theta)$ are generated and the corresponding $\tilde{A}_k$, $k = 1,...,B$ values of the test statistic are generated in the manner described above. The $p$-value of the null hypothesis that the observations are drawn from the specified distribution is finally computed as the fraction of $\tilde{A}_k$ values falling above $A$. Throughout this analysis, we use $M = 20$ and $B = 250$ (unless otherwise noted). This goodness-of-fit test assumes independence of the random draws $(w_{1n}, w_{2n}), (w_{1m}, w_{2m})$, $m \neq n$. To minimize the effect of serial dependence in data, in this study we subsample the datasets considered with a sampling interval sufficiently large to balance reducing serial dependence with maintaining sample size.





A second goodness of fit test proposed by McAssey (2013) was also considered, but the statistical power was found to be lower than the test described above for the distributions considered in this study.

*Acknowledgements.* This work was supported by the Natural Sciences and Engineering Research Council of Canada. The author gratefully acknowledges the provision of the 500 hPa data by the ECMWF, the tower data by the Cabauw Experimental Site for Atmospheric Research

5  (CESAR), and the sea surface wind data by the NASA Jet Propulsion Laboratory Physical Oceanography Distributed Active Archive Center. The author acknowledges helpful comments on the manuscript from Carsten Abraham and Arlan Dirkson.





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
