# Peer review of "Idealized Models of the Joint Probability Distribution of Wind Speeds"

_Nonlinear Processes in Geophysics, 2017_

## Referee Comment (RC1) · Anonymous Referee #1 · 18 Dec 2017

The manuscript is well structured, well written, and easy to follow. The contributions are clear.

The author makes use two joint PDFs, namely Weibull and Rice, to model wind speeds. These joint PDFs are validated and compared using field measurements. The conclusions are crystal clear. The author asserts that although Rice is more flexible than Weibull, the latter is mathematically more easily tractable. The author is unquestionably correct. On the other hand, I would like to call the attention of the author for another joint PDF that gathers a number of interesting features that may be useful for this application. This is the alpha-mu distribution [R1]. The alpha-mu joint PDF [R1, Eq. 28] contains the same number of parameters as the Rice one, encompasses Weibull as a special case (therefore, more flexible than it), and is mathematically as tractable

as Weibull. I believe the use of alpha-mu model will substantially increase the reference value of this manuscript. This is just a suggestion, which I will leave for the author and the Editor to decide to implement it or not.

On the cosmetic side, the author should have a look at the different font sizes appearing in Eqs. 17-21, 26-28, etc. (wherever the variable u_bar shows).

[1] M. D. Yacoub, The alpha-mu Distribution: A Physical Fading Model for the Stacy Distribution, IEEE Transactions on Vehicular Technology, Vol. 56, no. 1, pp. 27-34, January 2007.

---

## Referee Comment (RC2) · Anonymous Referee #2 · 19 Dec 2017

Summary: The article provides an empirical comparison of two statistical models for bivariate joint probability distribution of wind speeds. The bivariate Rice distribution is obtained by assuming that the projections of the two wind speeds on the X and Y axes are Gaussian and isotropic. The bivariate Weibull distribution arises from a non-linear transformation of components which are Gaussian, isotropic and centered. The comparison is performed in several different contexts: (i) 500hPa wind speeds at two different locations obtained from reanalysis data; (ii) 500hPa wind speeds at the same location at different moments in time, obtained from reanalysis data; (ii) observed wind speeds over land at the same location at 10m and 200m; (iv) see surface wind speeds at two different locations, obtained from satellite measurements. The two distributions are estimated by maximum likelihood, and statistical tests are performed to assess the

goodness of fit. The results are not completely unequivocal: for reanalysis data the bivariate Rice distribution seems to have a better performance, but the corresponding hypothesis is rejected in some statistical tests even for this law. For the measured land surface wind, neither of the two distributions fits the data well, but the performance improves for both laws when the data are conditionned on being in a specific wind regime.

Evaluation: both statistical distributions analysed in the present paper have already been published in the literature, and the main contribution of the present manuscript lies in the empirical analysis of their performance for wind speed modeling. The methodology of the paper is sound, and the results are well illustrated with data in several relevant settings. I see, however, several directions of improvement.

(i) The motivations of the paper are not fully clear. The analysis and the domain of applicability of the models, is limited to two dimensions. Where such two-dimensional wind speed models could be used? One possible use would be the vertical interpolation of wind speeds, with wind energy applications in view.

(ii) The conclusions are not crystal clear either. The author concludes that the Rice distribution is more flexible while the Weibull distribution is mathematically simpler and may be more convenient. Strictly speaking, for this conclusion one does not need the empirical analysis, they are clear simply by looking at the formulas. What precisely do we learn from the empirical study?

(iii) The message of the paper could be sharpened by considering more data. For example, (with the wind energy application in view) the goodness-of-fit for the wind speed distributions at different heights could be tested at many different locations, and the resulting p-values could be plotted on a map.

(iv) The interpretation of the results of the statistical tests could be improved. When the null hypothesis is rejected, is this due to the fact that the one-dimensional distributions do not fit the data well, or is it because the dependence structure is wrong? This

question could easily be answered by goodness-of-fit tests for the one-dimensional laws.

(v) A detailed comparison with other multivariate models could be performed. I do not fully agree with the author's statement that 'it is unlikely that a copula-based model will admit analytically tractable expressions'. Some copula families (Clayton, Gumbel etc.) are quite tractable, allow for multidimensional extensions, and their dependence structure, in particular, in the tails, is well understood. Another possibility would be to use a Gaussian dependence structure but apply a nonlinear transform to the components to produce positive wind speed values.

---

## Author Comment (AC1) · 6 Feb 2018

I have read and given great consideration to the reviews of my manuscript, "Idealized Models of the Joint Probability Distribution of Wind Speeds", NPG-2017-64, and I have modified the manuscript accordingly.

I would like to thank the Reviewers for their careful and helpful reviews of the manuscript. Following are my replies to the Reviewers' specific comments and a description of modifications to the manuscript.

**Reviewer 1**

I thank the reviewer for their positive review, and for drawing my attention to the $\alpha - \mu$ distribution. I have chosen not to add an analysis using this distribution to the revised

manuscript: as the Rice and Weibull distributions are standard parametric models for univariate wind speed distributions, it seems reasonable to me to focus on these distributions for an initial discussion of bivariate speed distributions. As well, adding a third bivariate speed distribution would make the already rather long manuscript even longer. I have added a discussion of this alternative bivariate distribution to the introduction (P3, LL 24-27) and conclusion (P24, LL 23-25), identifying consideration of this alternative bivariate speed distribution as an interesting direction of future research.

I also thank the reviewer for noting the typesetting issue they identify. I believe this can be addressed in the page proof stage. As the Journal likely has its in-house typesetting standards, it is not clear to me that this issue should be addressed at the stage of the revised submission.

**Reviewer 2**

I thank the reviewer for their detailed and constructive review, which has resulted in a number of modifications to the manuscript (detailed in the following)

1. *The motivations of the paper are not fully clear. The analysis and the domain of applicability of the models, is limited to two dimensions. Where such two-dimensional wind speed models could be used? One possible use would be the vertical interpolation of wind speeds, with wind energy applications in view.*

   I thank the reviewer for this comment. A discussion of the motivation of the study was present in the original manuscript, but was insufficiently emphasized. The manuscript has been revised as follows (P2, LL 16-22):

   Previous studies have used copula methods to model horizontal spatial dependence of wind speeds for wind power applications (Grothe and Schneiders, 2011; Louie, 2012; Veeramachaneni et al. 2015) and dependence of daily wind speed maxima (Schlözel and Friedrichs, 2008).
While these earlier analyses have focused on probabilistic modelling of simultaneous wind speed values at different spatial locations in the horizontal, dependence structures in the vertical (e.g. for vertical interpolation of wind speeds) or in time are also of interest. For example, an analysis in which the need for an explicit parametric form for the joint distribution of wind speeds at different altitudes has arisen is the Hidden Markov Model (HMM) analysis considered in Monahan et al. (2015).

2. *The conclusions are not crystal clear either. The author concludes that the Rice distribution is more flexible while the Weibull distribution is mathematically simpler and may be more convenient. Strictly speaking, for this conclusion one does not need the empirical analysis, they are clear simply by looking at the formulas. What precisely do we learn from the empirical study?*

I take the reviewer's point - the mathematical expression for the bivariate Weibull distribution is obviously simpler than that for the bivariate Rice. If the bivariate Weibull distribution was generically a good model for the joint distribution of speeds, then there would be no need to use the more complicated bivariate Rice. The empirical study allows us to assess the benefit that follows from using the more complicated Rice distribution (allowing e.g. representation of negatively correlated wind speeds). Furthermore, because wind speeds are neither Weibull nor Rice in truth, the empirical study allows assessment of the practical utility of the two bivariate models. The manuscript has been revised to make these points on Page 24, Lines 13-20 of the Conclusion:

> The fact that the bivariate Rice distribution is easier to work with, but less flexible, than the bivariate Weibull distribution is evident from inspection of their analytic forms and the relative number of parameters

to fit (5 vs. 6). If the bivariate Weibull distribution was generically appropriate for modelling the bivariate wind speed distribution, there would be no need to consider more complicated models such as the bivariate Rice. This study provides an empirical assessment of the relative practical utility of the two models, trading off the ability to model more general dependence structures (e.g. negatively correlated speeds) against model simplicity. Neither the univariate nor the bivariate Weibull or Rice distributions are expected to represent the true distributions of wind speeds (e.g. Carta et al., 2009). The results of this analysis characterize the practical utility of these models, rather than making a claim to their "truth".

3. *The message of the paper could be sharpened by considering more data. For example, (with the wind energy application in view) the goodness-of-fit for the wind speed distributions at different heights could be tested at many different locations, and the resulting p-values could be plotted on a map.*

I agree that such an analysis would be useful. However, tower stations are few with long (and freely available) records of wind speeds observed at altitudes sufficiently different for the wind speeds to show distinct variability. As a result, the spatial density of points in such an analysis would be quite low. Furthermore, this analysis would increase the length of an already long paper. I have chosen not to include any new datasets in the revised manuscript.

4. *The interpretation of the results of the statistical tests could be improved. When the null hypothesis is rejected, is this due to the fact that the one-dimensional distributions do not fit the data well, or is it because the dependence structure is wrong? This question could easily be answered by goodness-of-fit tests for the one-dimensional laws.*

A difficulty with separately testing the goodness-of-fit of univariate distributions is that the univariate test is not the same as the bivariate test, making direct comparison difficult. Even if the two tests have the same general form, the bivariate and univariate tests will not necessarily have the same power. A direct test to distinguish how well the bivariate distributions capture the dependence structure from how well they model the marginals is to repeat the bivariate goodness-of-fit test with one time series shuffled in time, thereby destroying the dependence structure. These analyses have been carried out and the results quoted in the revised manuscript. A description of this statistical test has been given on P. 14, LL 1-6 of the revised manuscript,

> In order to distinguish how well the parametric joint distributions model the marginal distributions from how well they represent dependence between variables, the goodness-of-fit analyses were repeated for each pair of time series with the values of one of the time series shuffled in time. This shuffling destroys the dependence structure without affecting the distributions of the marginals. Use of a bivariate analysis rather than separate univariate goodness-of-fit tests for the marginals allows direct comparisons of $p$-values, as exactly the same test is used for the original and shuffled data.

In general, we find that the $p$-values of fits to shuffled data are similar to or larger than those of fits to the original data: there is no general evidence of failure of the joint distributions to adequately model the data because of a failure to model the marginal distributions. These points are made for each of the datasets considered on P. 17, LL 10-14 and LL 27-30; P. 19 LL 19-22; and P. 21, LL 9-13. The following text has also been added to P. 24, LL 20-22 of the Conclusions:

> It is noteworthy that for the data considered in this study, the failure

of either the bivariate Rice or Weibull distributions to adequately fit the joint distribution of wind speeds (at a significance level of 5%) is not generally associated with a corresponding failure of the parametric distribution to model the marginals

5. *A detailed comparison with other multivariate models could be performed. I do not fully agree with the author's statement that 'it is unlikely that a copula-based model will admit analytically tractable expressions'. Some copula families (Clayton, Gumbel etc.) are quite tractable, allow for multidimensional extensions, and their dependence structure, in particular, in the tails, is well understood. Another possibility would be to use a Gaussian dependence structure but apply a nonlinear transform to the components to produce positive wind speed values.*

As noted in my response to Reviewer 1, adding consideration of other multivariate models would make the already rather long manuscript even longer. Consideration of other multivariate models as suggested by the reviewer has been identified as an interesting direction of future research in the revised manuscript (P. 24, LL 23-25)

I also note that my pessimistic statement about the analytic tractability of copula-based models applies to using copulas to characterize dependence in the vector components, rather than the speeds directly. It is the step of transforming to polar coordinates and integrating over angle to obtain a joint distribution for speeds that I expect to be analytically intractable.

A number of small changes of phrasing and grammar, have also been made to the revised manuscript. I sincerely hope that my responses to the reviewers' comments and the associated modifications to the manuscript are found to be satisfactory, and that the present revised manuscript is acceptable for publication in Nonlinear Processes in Geophysics.

Sincerely,

Adam Monahan